# Relationship between Physical Fitness and Match Performance Parameters of Chile Women’s National Football Team

**DOI:** 10.3390/ijerph18168412

**Published:** 2021-08-09

**Authors:** Rodrigo Villaseca-Vicuña, Fernando Manuel Otero-Saborido, Jorge Perez-Contreras, Jose Antonio Gonzalez-Jurado

**Affiliations:** 1Federación de Fútbol de Chile, 8320000 Santiago, Chile; 2Departamento de Educación Física, Deportes y Recreación, Universidad Metropolitana de Ciencias de la Educación, 8320000 Santiago, Chile; joperezc@gmail.com; 3Centro de investigación en Rendimiento Físico y Deportivo, Universidad Pablo de Olavide, 41001 Sevilla, Spain; fmotero@upo.es (F.M.O.-S.); jagonjur@upo.es (J.A.G.-J.)

**Keywords:** performance, training, soccer, women, competition, neuromuscular, aerobic, strength, resistance

## Abstract

The aim of this study was to analyze the relationships between the level of physical fitness and the physical performance parameters recorded by GPS in official FIFA matches of the Chilean women’s senior national football team in the period 2018–2020. Twenty-six female field players (age (mean ± SD) 26.8 ± 3.3 years, height 157.8 ± 21.5 cm, weight 58.9 ± 4.9 kg) participated in the study. Physical fitness variables were assessed: muscular strength, countermovement jump (CMJ), speed, agility and aerobic fitness. Physical performance variables were recorded by GPS in 26 official FIFA matches. The most notable associations with significant statistical significance (*p* < 0.001) were those observed between neuromuscular variables such as time to run 10 m (T10; *r* = −0.629) and jump (CMJ; *r* = 0.502) and the number of accelerations; aerobic fitness showed a highly significant relationship with meters run per minute (M/M; *r* = 0.589). The findings of this study provide evidence of how the level of physical fitness (neuromuscular and aerobic) relates to physical performance parameters recorded in official competitions.

## 1. Introduction

Football is the most played sport in the world, and although it has traditionally been played mainly by men, it is now played by both men and women [1,2]. The popularity of women’s football has grown in recent years. According to data published in 2019 by the International Federation of Association Football (FIFA), 73% of associations have an active senior women’s national team, a much higher ratio than the 55% recorded in 2015 [3]. Currently, there are around 13 million women playing organized football, and this figure is expected to increase to 60 million by 2026 [3]. In Chile, women’s football has undergone a progressive development, achieving an active participation of 9000 female players, 870 of whom are registered adult players and 870 of whom are registered players under the age of 18 [3]. Currently, the Chilean football federation has U-17, U-20 and Senior categories [3]. In this regard, research in women’s football has increased significantly over the last two decades [4,5]. Football professionals and sport scientists are looking for key factors that contribute to the optimization of elite female players’ performance both in training and in competition [6,7].

For example, the distance covered at different speeds at the Women’s World Cup in France 2019 has been recorded. The average total distance covered was 10,435 m, distributed in the following speed ranges: 0–7 km·h^−1^, 3735 m; 7–13 km·h^−1^, 3856 m; 13–19 km·h^−1^, 2168 m; 19–23 km·h^−1^, 495 m; and >23 km·h^−1^, 180 m [3]. Furthermore, players’ physical performance is assessed on a regular basis with the aim of identifying individual strengths and weaknesses, assessing the effects of training interventions, informing rehabilitation processes after injury and monitoring players’ development in competition [8]. In this sense, field tests are used to evaluate different physical qualities (strength, resistance, endurance, flexibility, power and speed) under conditions that are “ecologically” closer to the demands of actual competition [9]. Therefore, determining the associations between the fitness level and demands in official matches can help coaches, physical trainers and sport scientists to select the most valid tests to discriminate talent identification programs and optimize players’ performance [10].

There is no consensus in the literature on the relationship between the fitness level and performance in official matches. On the one hand, the aerobic fitness field test (level 1 recovery Yo-Yo) and neuromuscular performance tests have been found to be associated with physical performance variables obtained in matches such as total distance covered, distance covered at a high intensity and distance covered at a very high intensity [11,12]. Links have also been observed between the mean time between sprints in repeated sprint ability (RSA) tests and total and high-intensity distance covered during matches *(r* = −0.30 to 0.78) [10]. In this regard, Krustrup’s (2005) study showed in elite female players that the total distance covered and the distance covered at a high intensity in official matches are closely related to the fitness level [13]. In contrast, some research in adult and youth males has reported that certain fitness tests (CMJ, 5 m sprint time, aerobic fitness and RSA) are not related to competitive performance, nor to overall running performance during a football match [9,14]. Therefore, there are discrepancies in the scientific literature regarding the possible connections between the fitness level and the running performance in official matches, and it can be observed that there is scarce information available on the relationships between the fitness level and performance in official elite women’s football matches. Therefore, the aim of this study was to analyze the relationships between the fitness level and physical performance parameters recorded in official FIFA matches of the Chilean women’s senior national football team in the period 2018–2020.

## 2. Materials and Methods

### 2.1. Design

This is a quantitative, cross-sectional, descriptive, correlational and descriptive research work.

### 2.2. Participants

Female outfield players from the Chilean national football team participated in this study (*n* = 26; goalkeepers were not included). It was conducted in official FIFA and France 2019 World Cup matches in the period 2018–2020. The age (mean ± SD) was 26.8 ± 3.3 years, height was 157.8 ± 21.5 cm, weight was 58.9 ± 4.9 kg, muscle percentage was 48.28 ± 2.08% and fat percentage was 23.21 ± 2.03%. All the participants had played in a senior category in different leagues or professional competitions at the national level in different countries (Chile, Japan, Brazil, USA and Spain). They had 7 ± 2 years of experience in official competitions. At the time of the evaluations, the Chilean national team was ranked 36th out of 155 according to the FIFA women’s world ranking.

### 2.3. Timing

The players performed the fitness tests on the first 4 days of a weekly training cycle (Figure 1). All the players were required to avoid strenuous exercise 24 h prior to testing to prevent the effects of fatigue on the assessment.

The warm-up was led by the fitness coach. It consisted of a standardized general warm-up, which included gentle running, multidirectional movements and dynamic stretching, followed by a specific warm-up for each test. The total duration of the warm-up was 15–20 min.

Day 1, squat test (ST). Day 2, running speed, time to run 10 m (T10) and time to run 30 m (T30). Day 3, countermovement jump (CMJ) and Illinois agility time (IAT). Day 4, Yo-Yo intermittent recovery test level 1 (YYIRL1).

These tests are commonly performed by women’s football teams [1,5,6,15,16,17,18,19,20], as they provide valid data to assess the players’ physical fitness level and attributes, so all the participants were familiar with them and performed them regularly. Each player was encouraged to do each test with the maximum effort. GPS devices were used to record physical performance variables for competition in a total of 26 official matches (23 international friendlies and 3 corresponding to the France 2019 World Cup) (Figure 2). On official competition days, the players performed a standardized match warm-up that included general joint mobility exercises and specific exercises with the ball (passing, small-space games and defensive and offensive block exercises), with a total duration of 20–25 min. Players who participated more than 80 min in a match and who were free of any injury that would prevent maximum effort during the physical performance tests were included in the study.

### 2.4. Procedure

#### 2.4.1. Squat Test

This test has been validated to assess muscle strength levels in lower limbs in football players [21,22]. Before the measurement, each player performed a specific warm-up of 3 sets and 3 repetitions with a load of 20 kg. In the evaluation, each player performed 5 sets (1 set per load) of 3 repetitions, with loads of 20, 30, 40, 50 and 60 kg, respectively, with a recovery time of 3 min between sets. The concentric phase of the exercise was required to be performed at the maximum possible speed (Figure 3). No participant failed in any set. Two variables were recorded: (a) the relative strength (RS), obtained from the ratio of 1RM/body weight, and (b) the estimated 1RM, determined from the mean propulsive velocity (MPV) of the last load of the test (60 kg), calculated from the equation proposed by Pareja-Blanco, Walker and Häkkinen (2020) for the population of female athletes [22]. The aforementioned variables were determined using a linear encoder (Chronojump, Barcelona, Spain). The equation for calculating the 1RM in squat from the MPV at the last load is as follows:%1RM = (−42.196 MPV^2^ − 31.018 MPV + 112.937).

#### 2.4.2. Countermovement Jumping (CMJ)

This test is regularly performed to measure the lower extremity power of female football players [16,23]. Prior to testing, each player did a specific warm-up of 5 jumps to a jump box whose target surface was 40 cm above the ground. In the test, the participants made 3 attempts. The starting position is upright with hands on hips throughout the test to eliminate any arm swing influence. From this position the participant quickly flexes the knees to approximately 90 degrees and then immediately propels herself to jump vertically as high as possible, landing on both feet at the same time and with knees extended (approximately 180 degrees) [24]. If the evaluator observed an execution error, the jump was invalidated and the execution was repeated. This test was assessed using an Optojump Microgate (Bolzano, Italy) contact platform, with a 3 min rest between repetitions. The jump height (cm) was recorded and the best result was selected.

#### 2.4.3. Time in 10 (T10) and 30 m (T30)

This test is frequently used to assess acceleration and maximal running speed capacity in football players [15,25] (Figure 4). Prior to the assessment, each player performed a specific warm-up of 5 progressive 30 m sprints. There were 3 attempts of 30 m, with 3 min of recovery time between sprints. The test was carried out on a natural grass football pitch, in the morning at a temperature of 15 °C, with a relative humidity of 54%. The starting position was standing upright, placing the front foot just behind a line located 0.5 m away from the first photoelectric cell, to prevent the laser from being cut by the head or arms at the start of the run. The 3 photocells (Microgate, Bolzano, Italy) were placed at the start, at 10 m and at 30 m (Figure 4). The time of the three attempts was recorded at the following distances: 0–10 m (T10) as an indicator of acceleration and 0–30 m (T30) as an indicator of maximum speed.

#### 2.4.4. Illinois Agility Test

This test is often used to assess agility and changes of direction in football players [26,27]. Prior to measurement, each player performed a specific warm-up of 4 attempts at submaximal intensity. In the evaluation, the players performed 3 agility attempts with a recovery time of 3 min on the natural grass pitch. The shortest time of the 3 repetitions was taken. Photocells (Microgate, Bolzano, Italy) were used to measure the time taken for each attempt. The starting position was supine, with the feet placed 1 m behind the first beam (Figure 5). At the signal, the player covered the distance as fast as possible [28].

#### 2.4.5. Yo-Yo Intermittent Recovery Test Level 1 (YYIRL1)

This test is commonly used to assess intermittent endurance capacity in football players [29,30,31,32,33]. Prior to measurement, each player performed a specific warm-up consisting of 10 progressive 20 m runs with changes of direction. The players made one attempt at the test, following the test protocol [29]. The meters run (YYIRL1) were recorded for statistical analysis.

#### 2.4.6. GPS Monitoring

The total distance (TD) in meters, high-speed run >18 km·h^−1^ in meters (HSR), number of sprints at >18 km·h^−1^ (N°S), peak speed in km·h^−1^ (PS), meters per minute (M/M) and number of accelerations >2 m·s^−2^ (N°AC) were recorded. These data are often used by coaches and sport scientists to monitor loads and observe performance in training and official matches in women’s football [7,34,35,36,37,38,39]. Field players wore 10 Hz GPS devices (Optimeye S5, Catapult Sports, Melbourne, Australia) between the shoulder blades in a neoprene inner top.

### 2.5. Ethical Considerations

The study was conducted according to the guidelines of the Declaration of Helsinki (WMA, 2013) and approved by the Institutional Ethics Committee of University Hospitals Virgen Macarena and Virgen del Rocío from Seville, Spain (C.P. RENFEFUTCHILE—C.I. 2355-N-20, 28 June 2021). In addition, all the participants had a medical examination prior to the start of the season. All the participants underwent the tests without any injury or physical discomfort, and informed consent was obtained from all subjects involved in the study.

### 2.6. Statistical Analysis

For the descriptive analysis, the mean and standard deviation per playing position of the variables analyzed were calculated. The intraclass correlation coefficient (ICC) and coefficient of variation (CV) were found to analyze the relative and absolute reliability of variables that were measured more than twice. The Shapiro–Wilk test was performed to analyze whether the variables follow a normal distribution. Finally, correlations were calculated using Pearson or Spearman (*r*), according to normality, to study the level of association between physical fitness and physical performance variables in official matches. The *r* values were interpreted as trivial (0.00–0.09), small (0.10–0.29), moderate (0.30–0.49), large (0.50–0.69), very large (0.70–0.89), almost perfect (0.90–0.99) and perfect (1.0) [40]. The 95% confidence interval (CI) was calculated for all the measures. Statistical significance was established for a value of *p* ≤ 0.05. Statistical analysis was carried out using SPSS IBM software, Version 22 (New York, NY, USA).

## 3. Results

Table 1 shows the physical fitness variables of the field players.

Table 2 shows the variables recorded with GPS in official matches based on the FIFA calendar (2018–2020).

Table 3 shows the relationships between the physical fitness variables of neuromuscular performance and performance parameters recorded in official matches. A direct and significant relationship was recorded between the number of accelerations (N°AC) and the relative strength and the CMJ, while the N°AC was inversely related to the 10 and 30 m running speed, as well as the results in the agility test.

Figure 6 shows the simple linear regressions in which statistically significant associations were found between neuromuscular variables and physical performance variables in official matches. The most notable inverse relationship was found between the number of accelerations and the time to run 10 m, and the most significant positive association was observed between the number of accelerations and the CMJ.

Figure 7 shows the relationships between intermittent aerobic fitness (YYIRL1) and physical performance variables in official matches. All the associations shown are direct and statistically significant, although the most remarkable is the one registered between YYIRL1 and the meters run per minute in a match.

## 4. Discussion

The aim of this study was to analyze the relationships between the level of physical fitness and physical performance parameters recorded in official FIFA matches of the Chilean women’s senior national football team in the period 2018–2020.

Significant relationships were found between neuromuscular performance-oriented fitness variables and physical performance indicators in official matches (Table 3 and Figure 7): a significant association was observed between 1RM and TD (*r* = −0.398), and RS, T10, T30, CMJ and IAT were observed to be statistically significantly associated with N°AC (*r* = 0.491, −0.629, −0.485, 0.502 and −0.422, respectively). These results are in agreement with those reported by Rago et al. [14], who investigated the relationships between the fitness level and the activity pattern in small-sided games and matches with regulation dimensions in a sample of fourteen Italian elite male senior football players. These authors reported significant relationships of CMJ scores with TD (*r* = 0.40) and with N°AC (*r* = 0.57). Similar associations were found between 5-m sprints and the N°AC in small-sided games and in regulation dimensions (*r* = −0.48 and −0.79, respectively). These results might suggest that players who are able to produce a lot of force per unit time in the explosive tests of sprinting (5 m sprint) and jumping (CMJ) will be able to perform more high-intensity accelerations in training and in football matches. The findings of Rago et al. [14] coincide with the findings of this study; i.e., the higher the speed in movements (shorter T10 and T30 times), the higher the number of accelerations. Although there is an explained variance of 40–23% between T10–T30 and N°AC, we cannot establish a cause-and-effect relationship. This could be explained by the fact that players with greater strength may have a greater capacity to transfer to high-intensity actions in real competitive actions [41], and it has also been demonstrated that RS is a variable that is related to neuromuscular performance variables in elite women; this could have an explanation, given that the players who manifest greater relative strength are capable of producing greater force in relation to their body weight, understanding that most actions in soccer are precisely mobilizing the same body weight of the player in specific actions (jumps, accelerations, sprinting) [42]. It has also been shown that this indicator in low performance could be related to the risk of injury in athletes [43]. Therefore, assessing RS could be a discriminatory variable in elite women’s soccer performance. However, these results do not fully agree with those reported by Aquino et al. [9], who studied the relationships between field testing and running performance in football matches using computer tracking technology in young Brazilian elite football players. Their results showed no relationships between running tests (T10 and T30) and performance in official matches. However, the authors reported that the agility test (zigzag) explained 17% of the total variance in PS during real matches. These differences could be explained by Aquino et al. [9] using a sample of young players. This population may possibly not be able to express their maximum neuromuscular fitness potential according to the demands in official matches. Furthermore, match performance is also influenced by other variables (effects of altitude, temperature, match outcome, opponent’s competitive level, match schedules and competitive calendar congestion) [42].

On the other hand, the aerobic performance-oriented fitness variable YYIRL1 (Figure 7) showed relationships with TD, HSR, N°S and M/M (*r* = 0.490, 0.390, 0.491 and 0.589, respectively). These results are in agreement with those reported by Rago et al. [14] who investigated the relationships between the level of aerobic fitness and the activity pattern in small-sided games and regulation dimension (full-size) football matches in elite Italian male footballers. The results showed that YYIRL1 was related to TD in small-sided games (*r* = 0.47) and full-size football (*r* = 0.57) and also to HSR in full-size football (*r* = 0.46). On the other hand, a negative correlation was observed with N°AC (*r* = 0.49; *p* < 0.05). Something similar was reported by Aquino et al. [9], who studied the relationships between field testing and running performance in official matches using computational tracking technology in elite Brazilian youth football players. The results showed that YYIRL1 was positively correlated with the V4 speed range (medium-intensity running). Similarly, Castagna, Manzi, Impellizzeri, Weston and Barbero [12] studied the relationship between YYIRL1 and match physical performance in elite male youth football players, and their results showed significant relationships with HSR (*r* = 0.76). Krustrup et al. [13], who assessed aerobic performance using YYIRL1, found significant relationships with TD (*r* = 0.56) and HSR (0.76). These findings suggest that the YYIRL1 test has a high sensitivity to the ability to run longer distances and at a higher intensity in official football matches, irrespective of the level or sex of the football players. For example, young football players who performed better on the YYIRL1 test covered a greater total distance (*r* = 0.65) and ran more at higher intensities (*r* = 0.77) [43]. Specifically, performance on the YYIRL1 test was associated with high-intensity running (13.0–18.0 km·h^−1^; *R*^2^ = 42%) and sprints (˃18.0 km·h^−1^; *R*^2^ = 57%) [12]. In male professional players, match running performance was also associated with different variables assessed by field tests. Krustrup et al. [33] reported correlations between YYIRL1 and total distance run (*R*^2^ = 28%), high-intensity running (15–18 km·h^−1^; *R*^2^ = 50%) and total high-intensity activities (˃18 km·h^−1^; *R*^2^ = 33%). In professional female football players, YYIRL1 scores were related to total distance covered (*r* = 0.56; *R*^2^ = 31%) and distance covered in high-intensity running during matches (∑0−90 min, *r* = 0.76, *R*^2^ = 57%; ∑30–45 and 75–90 min, *r* = 0.83, *R*^2^ = 69%) [13].

Finally, the findings presented in our study demonstrate strong associations between YYIRL1 and distances covered in official matches. Furthermore, YYIRL1 is one of the few field tests that demonstrates a similarity in the external load represented by distance run per minute (M/M) in official matches. For example, taking our records, the mean duration of YYIRL1 is 12 min and the mean distance covered is 1436 m. Therefore, the estimated work rate is 118 M/M (Table 1), and that observed during official football matches is 108 M/M (Table 2). This could suggest a strong association between the demands represented in this field test and official matches [29,31,33], which could provide valuable information to differentiate, by means of field tests, the ability to cover greater distance and run at higher speed in official matches.

## 5. Conclusions

The main findings of this study provide evidence on how the level of physical fitness (neuromuscular and aerobic) is related to physical performance parameters recorded in official matches using GPS devices. On the one hand, neuromuscular fitness indicators could be good predictors of N°AC in official matches, particularly performance in the T10 and CMJ tests, explaining more variance in the number of high-intensity accelerations in official competitions. On the other hand, the intermittent endurance capacity obtained in the YYIRL1 test is a good predictor of performance in total distance covered (TD) and distances covered at a high speed in elite women’s football matches. These findings therefore suggest that analyzing the level of physical fitness by means of field tests could provide relevant information to select players with a greater aptitude for elite competition.

From the results obtained, we can conclude that there is a moderate relationship between 1RM and TD, and RS, T10, T30, CMJ and IAT exhibit moderate relationships with N°AC. On the one hand, neuromuscular fitness indicators could explain the performance of N°AC in official matches and in particular of the performance in T10 and CMJ tests, according to a greater variance in the number of high-intensity accelerations in official competitions. In addition, the intermittent endurance capacity obtained in the YYIRL1 test may be a good parameter of performance in total distance run (TD) and distances run at high intensity in elite women’s soccer matches. These results suggest, therefore, that the analysis of the level of physical fitness through field tests can be a useful indicator to identify those players who present a better sporting form to compete in high performance; however, by obtaining moderate relationships, they do not allow prediction of performance, considering also that field tests cannot be used as predictors of performance due to the complex and multifactorial nature of the competition [44], but the data obtained in this study can be used for reference and comparative analysis, as well as to evaluate specific physical components of the sport specialty and to individualize training loads.

### Practical Applications

Field evaluations represent an interesting approach to assess the physical fitness of athletes and thus identify which ones are in a better sporting shape to withstand the demands of high-performance competition. The results of our study demonstrate that fitness levels (neuromuscular and aerobic) are associated with measures of competitive performance.

## Figures and Tables

**Figure 1 ijerph-18-08412-f001:**
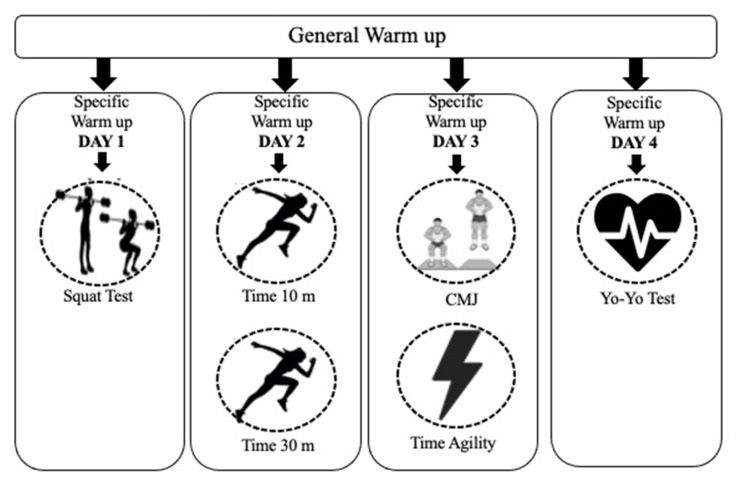
Physical fitness test timeline.

**Figure 2 ijerph-18-08412-f002:**
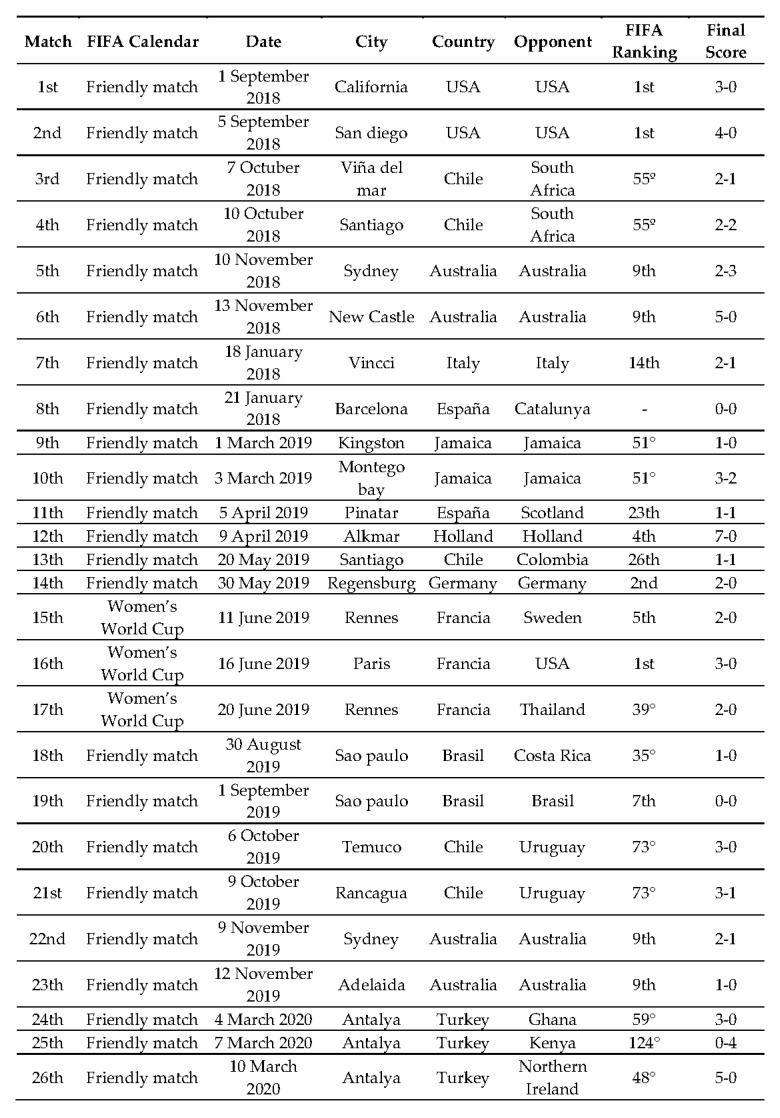
Official FIFA and France 2019 World Cup match schedule analyzed for the study.

**Figure 3 ijerph-18-08412-f003:**
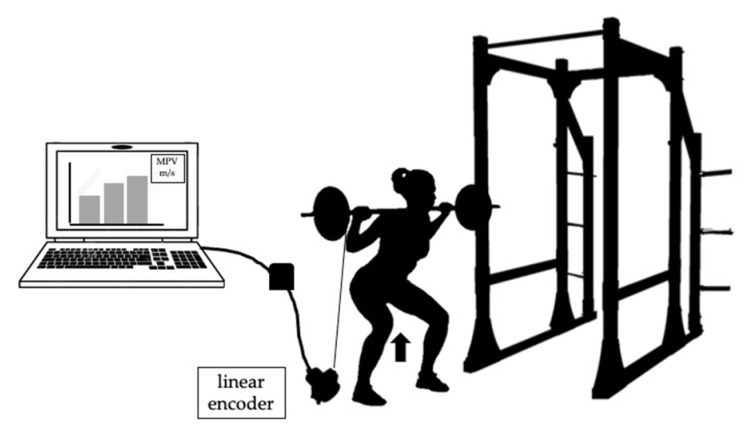
Squat test.

**Figure 4 ijerph-18-08412-f004:**
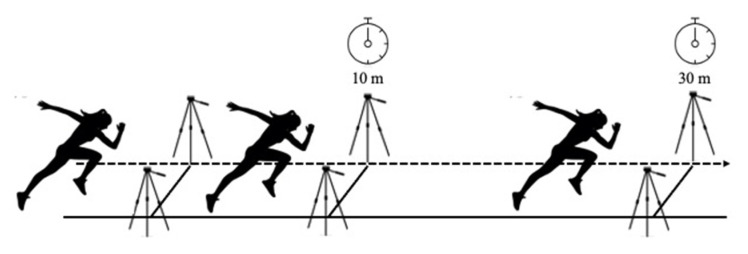
Acceleration evaluations at 10 m and velocity at 30 m.

**Figure 5 ijerph-18-08412-f005:**
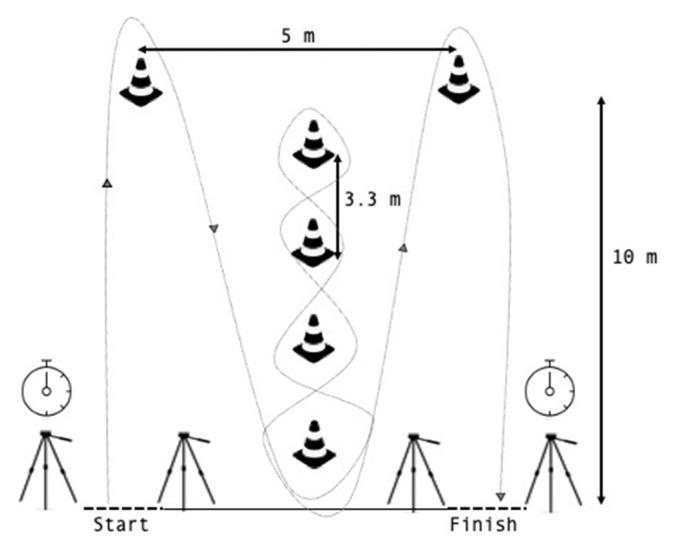
Illinois agility test.

**Figure 6 ijerph-18-08412-f006:**
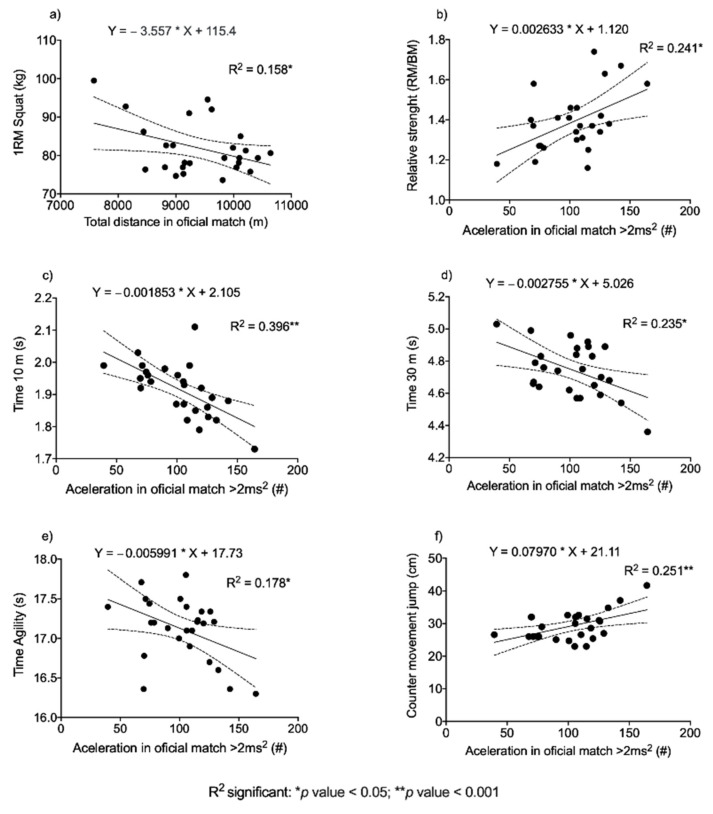
Simple linear regressions between neuromuscular variables and official match performance. (**a**) 1RM squat with Total distance; (**b**) Relative Strenght with Aceleration; (**c**) Time 10 m with Aceleration; (**d**) Time 30 m with Aceleration; (**e**) Time Agility with Aceleration; (**f**) Counter movement jump with Aceleration.

**Figure 7 ijerph-18-08412-f007:**
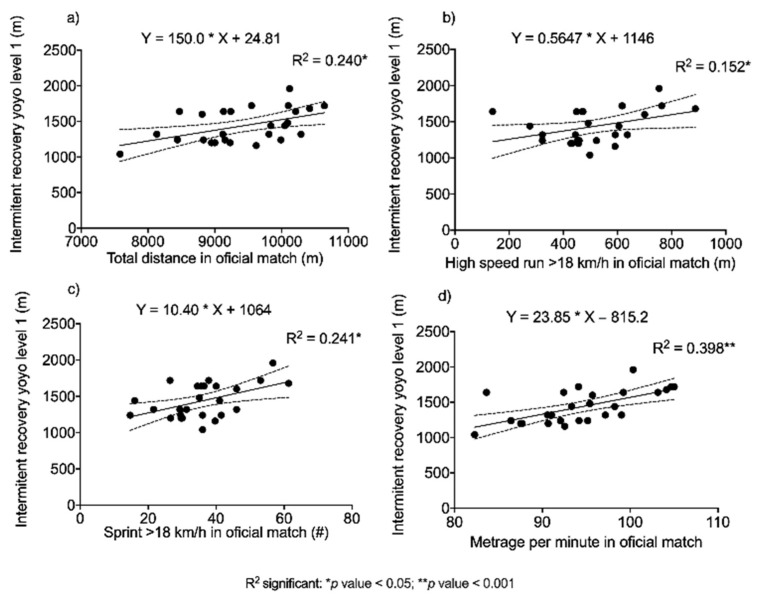
Simple linear regressions between aerobic capacity and official match performance. (**a**) Total distance; (**b**) High speed run; (**c**) Number sprint; (**d**) Metrage per minute.

**Table 1 ijerph-18-08412-t001:** Physical fitness variables (*n* = 26).

Variables	M ± SD	CI (95%)	ICC	CV (%)
1RM (kg)	81.87 ± 6.85	(79.10–84.64)	-	-
RS (1RM/BW)	1.38 ± 0.15	(1.32–1.45)	-	-
CMJ (cm)	29.26 ± 4.45	(27.46–31.06)	0.99	15.2
T10 (s)	1.9 ± 0.08	(1.88–1.94)	0.97	4.31
T30 (s)	4.74 ± 0.15	(4.67–4.80)	0.56	3.36
IAT (s)	17.11 ± 0.39	(16.95–17.27)	0.93	2.32
YYIRL1 (m)	1436.92 ± 234.85	(1342.92–1531.78)	-	-

Mean ± standard deviation (M ± SD); confidence interval (CI) of 95%. 1RM: 1 repetition maximum squat; RS: relative strength; T10: time 10 m; T30: time 30 m; IAT: Illinois agility time; CMJ: countermovement jump. YYIRL1: meters in Yo-Yo test.

**Table 2 ijerph-18-08412-t002:** Variables obtained from GPS in official matches (*n* = 26).

Variables	M ± SD	CI (95%)
TD (m)	9415.16 ± 766.69	(9105.48–9724.82)
HSR (m)	515.34 ± 162.31	(449.78–580.90)
N°S	35.28 ± 11.09	(31.34–40.31)
PS (km·h^−1^)	25.27 ± 2.65	(24.02–26.34)
M/M	108.14 ± 6.97	(105.32–110.96)
N°AC > 2 m·s^−2^	102.36 ± 28.04	(91.03–113.69)

Mean ± standard deviation (M ± SD) and confidence interval (CI) of 95%. TD: total distance; HSR: high-speed running (>18 km·h^−1^); N°S: number of sprints; PS: peak speed; M/M: meters per minute; N°AC: number of accelerations.

**Table 3 ijerph-18-08412-t003:** Correlations between physical fitness and performance parameters in official matches.

		Match Performance					
		TD	HSR	N°S	PS	M/M	N°AC
Neuromuscular Performance	1RM	−0.398 *	0.053	−0.054	0.304	−0.306	0.256
	RS	−0.07	0.177	0.271	0.097	−0.021	0.491 *
	T10	0.21	−0.159	−0.329	0.017	0.124	−0.629 **
	T30	0.255	−0.259	−0.191	−0.291	0.173	−0.485 *
	CMJ	−0.271	0.157	0.031	0.296	−0.207	0.502 **
	IAT	0.328	−0.091	0.044	−0.333	0.26	−0.422 *
Aerobic Power	YYIRL1	0.490 *	0.390 *	0.491 *	−0.153	0.589 **	0.188

** Correlation is significant at the 0.01 level (bilateral). * Correlation is significant at the 0.05 level (bilateral). 1RM: 1 repetition maximum squat; RS: relative strength; T10: time 10 m; T30: time 30 m; IAT: Illinois agility time; CMJ: countermovement jump; YYIRL1: meters in Yo-Yo test; TD: total distance; HSR: high-speed running; N°S: number of sprints; PS: peak speed; M/M: meters per minute; N°AC: number of accelerations.

## Data Availability

The data presented in this study are available on request from the corresponding author. The data are not publicly available due to privacy.

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
