# Peer review of "Relationship between Physical Fitness and Match Performance Parameters of Chile Women’s National Football Team"

_ijerph, 2021, doi:10.3390/ijerph18168412_

Round 1
Reviewer 1 Report
In the theoretical approach, clearly separate when the bibliographic references in the theoretical framework are to studies with female or male football players. If there are no studies with girls and other studies with boys are used as a reference, this should be explicitly highlighted.
It could have been considered to maintain 48 hours without strenuous exercise instead of 24 hours before the tests (Recovery period for some strength work).
Review the translation. For example, "30 metros" appears in spanish (section 2.4.3.).
Author Response
-In the theoretical approach, clearly separate when the bibliographic references in the theoretical framework are to studies with female or male football players. If there are no studies with girls and other studies with boys are used as a reference, this should be explicitly highlighted.
RESPONSE: Thank you, your recommendation has already been improved in the study.
-It could have been considered to maintain 48 hours without strenuous exercise instead of 24 hours before the tests (Recovery period for some strength work).
RESPONSE: Thank you for your comments, we will take it into consideration for a future study.
-Review the translation. For example, "30 metros" appears in spanish (section 2.4.3.).
RESPONSE: Thank you, your recommendation has already been changed "metros" to "metres"

Reviewer 2 Report
Relationship between physical condition and performance parameters in official matches of Chile’s national women’s football team female.
Major concerns
- there is no mentioned in the method section that this study has received either an official ethical clearance from any human research ethics board nor did the study had informed written permission from the all players involved. This should be clearly stated or authors must provide a very good justification why is such the case. The current sentence written, “This type of intervention does not alter normal football training, nor does it involve motor actions different from normal training or match practice” is not acceptable as ethical clearance nor serve as informed consent from the players.
- while there are statistically significant correlations between some of the physical fitness measures and the some of the match performance parameters, the variance of these variables were generally low. For example, the variance between the highest correlations, i.e., T10 and number of accelerations is only 40%. So, it is important for the authors to highlight this point to the readers that correlations are not cause-and-effect and given the low variance between some of these variables, readers need to take caution when interpreting these relationships.
- the high number of friendly matches relative to actual competitive matches. Because the intensity of match could be very different between friendly and competitive matches, I would suggest that authors perform a statistical analysis to compare the overall players’ match performance during these 2 different match-settings. And only if they find no statistical difference in the players’ match performance (i.e., distance covered, high-intensity runs, etc) between these 2 match settings, can the 23 friendly and 3 actual official competitive matches be combined. Alternatively, a separate correlation analyses could be done between the match performance parameters with the 23 friendly matches and with the 3 official competitive matches, separately – and then see whether there are any differences in the values of the corelations obtained.
- In the introduction/discussion section, there should be a paragraph stating why the authors would expect the relationship between physical condition and match related performance in Women to be different from men. As an example, the authors could argue that women use more fats during exercise and thus their pattern of movement is more aerobically predominant than men and thus female soccer players would be expected to cover a higher percentage of their match performance in the low and moderate intensity range (or speeds) and cover less high-intensity running. As another example, because female has less strength than men, the female game may probably be more on the ground passes to cover the field and then you would expect the female players to run/walk or cover more distance relative to men. In short, please provide a rationale to doing this study – because many studies have been done on men – and why would you expect a different result when doing the same study with women; if yes there is difference why and if there is no difference between men and women why not as well.
Minor issues
- title. Change “condition” to “fitness” and use the term “match performance parameters”
- Line 57 to 70. It should be clearly stated that the discussions on the relationship between the various physical attributes and match performances were conducted in MALE elite players. The current discussion did not indicate that this was the case.
- Like 82. Write out the number 26.
- Like 88. “… experience in official competitions”. This is not clear – is this league or international matches. I think authors should detail a bit more the experience of the players involved in the study. Years of training and playing experience at the club and international levels. Are they professional or amateur levels?
- Figure 4 and Figure 5 are common tests and well-known tests and hence not necessary. Please delete. Similar for Figure 7 and Figure 8 where the same information has been reported in Table 3 and does not add any new information.
- Line 106. “….. to assess the players’ physical fitness and attributes …..”
- Line 150. “metres” not “metros”.
- Line 154. ‘’… recovery time between sprints”.
- Line 166. Delete “(TAGI)”.
- Line 184. “GPS monitoring The Total distance (TD), High Speed Run >18km/h in metres (HSR), number of sprints at >18km/h (NºS), Peak Velocity in km/h (PV), Metre per minute (M/M), and Number of Accelerations >2ms2 (NºAC) were recorded”. Change all these variables to “total distance (TD in metres, m) during match, High Speed Run (HSR in m) >18.0 km·h-1, number of sprints at >18.0 km·h-1, peak velocity (PV in km·h-1), work rate (in m·min-1), and number of accelerations >2.0 m·s-2 were recorded”
- Is it peak velocity or peak speed? – please be consistent throughout the manuscript
- Table 1 and Table 2. Write out the variables.
- Figure 2. Please delete all the small figures of calendar, house, balls and cup; they do not add value to the figure. Also, delete the Chile in every box because all matches are against Chile. The dates should be written as an example, date-month, like 09 Sep to be clearer to reader. I think it would be useful to write the match scores to provide some indication of the “intensity” of the matches.
- The Yo-Yo Intermittent Recovery test level 1 should be shortened as YYIRL1 and not MYYRI.
- Line 242 onwards. In the discussion section, the authors should try to provide the reason(s) for the positive correlation between Strength (lower limb) and T10 with acceleration during matches. Could it be that the players are who felt that they are stronger and speedier have a greater desire to perform accelerations during matches because they are good in it (since they possess the strength and the speed to move). The same point for the observed positive relationships between aerobic fitness and several other match performance parameters. It is important that throughout the discussion section that when the authors highlighting the previous research to the reader whether the studies cited were conducted in male or female players – to make it clear to the reader. For example, in the cited Rago et al paper – was the study conducted in male or female footballers to suggest similar findings to the present study?
Author Response
REVIEWER 2
Major concerns
- there is no mentioned in the method section that this study has received either an official ethical clearance from any human research ethics board nor did the study had informed written permission from the all players involved. This should be clearly stated or authors must provide a very good justification why is such the case. The current sentence written, “This type of intervention does not alter normal football training, nor does it involve motor actions different from normal training or match practice” is not acceptable as ethical clearance nor serve as informed consent from the players.
RESPONSE: Thank you very much for your comment. Corrected lines 209-215, and we will attach the approval of the ethics committee.
- while there are statistically significant correlations between some of the physical fitness measures and the some of the match performance parameters, the variance of these variables were generally low. For example, the variance between the highest correlations, i.e., T10 and number of accelerations is only 40%. So, it is important for the authors to highlight this point to the readers that correlations are not cause-and-effect and given the low variance between some of these variables, readers need to take caution when interpreting these relationships.
RESPONSE: Thank you very much for your comment, we will incorporate the phrase in lines 312-315 “The findings of Rago et al. [14] coincide with the findings of this study, ie, the higher the speed in movements (the shorter time at T10 and T30), the higher the number of accelerations, although there is an explained variance of 40-23% between T10- T30 and Nº AC, we cannot establish a cause‐and‐effect relationship "
- the high number of friendly matches relative to actual competitive matches. Because the intensity of match could be very different between friendly and competitive matches, I would suggest that authors perform a statistical analysis to compare the overall players’ match performance during these 2 different match-settings. And only if they find no statistical difference in the players’ match performance (i.e., distance covered, high-intensity runs, etc) between these 2 match settings, can the 23 friendly and 3 actual official competitive matches be combined. Alternatively, a separate correlation analyses could be done between the match performance parameters with the 23 friendly matches and with the 3 official competitive matches, separately – and then see whether there are any differences in the values of the corelations obtained.
RESPONSE: Thanks for your suggestions. We have already done this analysis, and the results showed that the external load of international friendly matches on FIFA date (n=23) does not differ from the women's World Cup matches (n = 3). Our group is working on this hypothesis, but we are going to collect more data from official matches because the sample is low, just n=3. For example right now from Olympic Games in Japan, where the Chile women’s national football team is.
- In the introduction/discussion section, there should be a paragraph stating why the authors would expect the relationship between physical condition and match related performance in Women to be different from men. As an example, the authors could argue that women use more fats during exercise and thus their pattern of movement is more aerobically predominant than men and thus female soccer players would be expected to cover a higher percentage of their match performance in the low and moderate intensity range (or speeds) and cover less high-intensity running. As another example, because female has less strength than men, the female game may probably be more on the ground passes to cover the field and then you would expect the female players to run/walk or cover more distance relative to men. In short, please provide a rationale to doing this study – because many studies have been done on men – and why would you expect a different result when doing the same study with women; if yes there is difference why and if there is no difference between men and women why not as well.
RESPONSE: Thank you for your comments, we think are very interesting, but this study is focused on female football. In fact, the objective is to analyse the relationships between the fitness level and physical performance parameters recorded in women football players... There are a huge number of papers comparing females vs. males, very interesting, but it is not the goal in this study.
Although it is true that in the discussion, we have cited some investigations developed with males, it was because there are few studies about this subject in women. If we establish in the introduction the comparison “males vs females" as support to this paper, we think that we are going to pervert this research. We are convinced that women must be the protagonists in this research, and we would like it to be possible.
MINOR ISSUES
- title. Change “condition” to “fitness” and use the term “match performance parameters”
RESPONSE: Thank you for your comment, this suggestion has already been improved in our study.
- Line 57 to 70. It should be clearly stated that the discussions on the relationship between the various physical attributes and match performances were conducted in MALE elite players. The current discussion did not indicate that this was the case.
RESPONSE: Thank you for your comment, this suggestion has already been improved in our study specifically in line 71-76; "In this regard, Krustrup's (2005) study showed in elite female players that the total dis-tance covered and the distance covered at a high intensity in official matches is closely related to the fitness level [13]. In contrast, some research in adult and youth males has reported that certain fitness tests (CMJ, 5 m sprint time, aerobic fitness and RSA) are not related to competitive performance, nor to overall running performance during a football match [9,14]. "
- Like 82. Write out the number 26.
Thank you. Corrected
- Like 88. “… experience in official competitions”. This is not clear – is this league or international matches. I think authors should detail a bit more the experience of the players involved in the study. Years of training and playing experience at the club and international levels. Are they professional or amateur levels?
RESPONSE: Thank you for your comments, this point will be improved. However, it appears clearly in point 2.2 participants, that the players are from the Chilean national team, and that they participate in the professional leagues of their clubs in different countries. It also states “they had 7 ± 2 years of experience in official competitions. At the time of the evaluations, the Chilean national team was ranked 36th out of 155 according to the FIFA women's world ranking. ”
- Figure 4 and Figure 5 are common tests and well-known tests and hence not necessary. Please delete.
RESPONSE: In previous submits to MDPI journals, Editors ask to authors, if it is possible, to include figures or graphics to facilitate the reader's comprehension. We would like to achieve the widest possible dissemination in this research. Not just for expert scientists, but also for coaches, fitness coaches, or beginner scientists. In addition, in our opinion (like editor's opinion) this graphics or figures make an easier and more enjoyable reading. Please, could you require to editors about this issue? Thank you in advance.
--Similar for Figure 7 and Figure 8 where the same information has been reported in Table 3 and does not add any new information.
RESPONSE: Figures 7 and 8 show different information than Table 3.
Table 3 shows the results of Correlations between physical condition and performance parameters, meaning, r of Pearson, and p-value.
In Figures 7 and 8 we can see the regression lines with their confidence interval (showing how is the relation between both variables) the regression equations (you can estimate the value of the variable Y knowing the value of the variable X) and the R2 ( explained variance)
- Line 106. “….. to assess the players’ physical fitness and attributes …..”
RESPONSE: thank you for your comment, this was improved in line 116 "... to assess the players physical fitness level and attributes, so all the participants were ..."
- Line 150. “metres” not “metros”.
RESPONSE: Thank you, your recommendation has already been changed "metros" to "metres"
- Line 154. ‘’… recovery time between sprints”.
RESPONSE: Thanks for your comment, this was improved on line 169 ". There were 3 attempts of 30 m, with 3 minutes of recovery time between sprints."
- Line 166. Delete “(TAGI)”.
RESPONSE: Thank you for your comments, we will include the acronym “IAT” which refers to “Illinois Agility Time”. It is precisely the variable we use, the time it takes to finish the agility test.
- Line 184. “GPS monitoring The Total distance (TD), High Speed Run >18km/h in metres (HSR), number of sprints at >18km/h (NºS), Peak Velocity in km/h (PV), Metre per minute (M/M), and Number of Accelerations >2ms2 (NºAC) were recorded”. Change all these variables to “total distance (TD in metres, m) during match, High Speed Run (HSR in m) >18.0 km·h-1, number of sprints at >18.0 km·h-1, peak velocity (PV in km·h-1), work rate (in m·min-1), and number of accelerations >2.0 m·s-2 were recorded”
RESPONSE: Thanks for your comments, these were improved in the article
- Is it peak velocity or peak speed? – please be consistent throughout the manuscript
RESPONSE: Thanks for your comment, this was improved, and we will use Peak Speed (PS).
- Table 1 and Table 2. Write out the variables.
RESPONSE: Thanks for your comment. Corrected.
- Figure 2. Please delete all the small figures of calendar, house, balls and cup; they do not add value to the figure. Also, delete the Chile in every box because all matches are against Chile. The dates should be written as an example, date-month, like 09 Sep to be clearer to reader. I think it would be useful to write the match scores to provide some indication of the “intensity” of the matches.
RESPONSE: Thanks for your comment. Corrected.
- The Yo-Yo Intermittent Recovery test level 1 should be shortened as YYIRL1 and not MYYRI.
RESPONSE: Thanks for your comment, this was improved and replaced throughout the study "MYYRI" by "YYIRL1"
- Line 242 onwards. In the discussion section, the authors should try to provide the reason(s) for the positive correlation between Strength (lower limb) and T10 with acceleration during matches. Could it be that the players are who felt that they are stronger and speedier have a greater desire to perform accelerations during matches because they are good in it (since they possess the strength and the speed to move). The same point for the observed positive relationships between aerobic fitness and several other match performance parameters. It is important that throughout the discussion section that when the authors highlighting the previous research to the reader whether the studies cited were conducted in male or female players – to make it clear to the reader. For example, in the cited Rago et al paper – was the study conducted in male or female footballers to suggest similar findings to the present study?
RESPONSE: Thank you for your comments, this point will be improved. We add in line 285-294 “This could be explained by the fact that players with greater strength may have a greater capacity to transfer to high-intensity actions in real competitive actions [41], and it has also been demonstrated that rs is a variable that is related to performance variables neuromuscular in female elite women, this could have an explanation, given that the players who manifest greater relative strength are capable of producing greater force in relation to their body weight, understanding that most actions in soccer are precisely mobilizing the same body weight of the player in specific actions (jumps, accelerations, sprinting) [42], it has also been shown that this indicator in low performance could be related to the risk of injury in athletes [43]. Therefore, assessing RS could be a discriminatory variable in elite women's soccer performance. " And we also add in rago et al. who were "fourteen Italian elite male senior football players"

Round 2
Reviewer 2 Report
none.